# META ADVERSARIAL TRAINING

## ABSTRACT

Recently demonstrated physical-world adversarial attacks have exposed vulnerabilities in perception systems that pose severe risks for safety-critical applications such as autonomous driving. These attacks place adversarial artifacts in the physical world that indirectly cause the addition of universal perturbations to inputs of a model that can fool it in a variety of contexts. Adversarial training is the most effective defense against image-dependent adversarial attacks. However, tailoring adversarial training to universal perturbations is computationally expensive since the optimal universal perturbations depend on the model weights which change during training. We propose meta adversarial training (MAT), a novel combination of adversarial training with meta-learning, which overcomes this challenge by meta-learning universal perturbations along with model training. MAT requires little extra computation while continuously adapting a large set of perturbations to the current model. We present results for universal patch and universal perturbation attacks on image classification and traffic-light detection. MAT considerably increases robustness against universal patch attacks compared to prior work.

## 1 INTRODUCTION

Deep learning is currently the most promising method for open-world perception tasks such as in automated driving and robotics. However, the use in safety-critical domains is questionable, since a lack of robustness of deep learning-based perception has been demonstrated (Szegedy et al., 2014; Goodfellow et al., 2015; Metzen et al., 2017; Hendrycks & Dietterich, 2019).

Physical-world adversarial attacks (Kurakin et al., 2017; Athalye et al., 2018; Braunegg et al., 2020) are one of most problematic failures in robustness of deep learning. Examples of such attacks are fooling models for traffic sign recognition (Chen et al., 2018; Eykholt et al., 2018a;b; Huang et al., 2019), face recognition (Sharif et al., 2016; 2017), optical flow estimation (Ranjan et al., 2019), person detection (Thys et al., 2019; Wu et al., 2020b; Xu et al., 2020), and LiDAR perception (Cao et al., 2019). In this work, we focus on two subsets of these physical-world attacks: local ones which place a printed pattern in a scene that does not overlap with the target object (Lee & Kolter, 2019; Huang et al., 2019) and global ones which attach a mainly-translucent sticker on the lens of a camera (Li et al., 2019). Note that these physical-world attacks have cor-

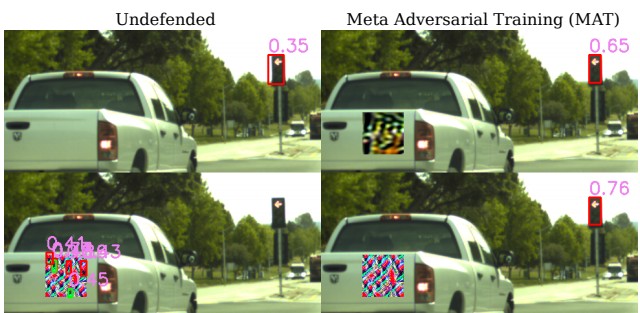

Figure 1: Illustration of a digital universal patch attack against an undefended model (left) and a model defended with meta adversarial training (MAT, right) on Bosch Small Traffic Lights (Behrendt & Novak, 2017). A patch can lead the undefended model to detect non-existent traffic lights and miss real ones that would be detected without the patch (bottom left). In contrast, the same patch is ineffective against a MAT model (bottom right). Moreover, a patch optimized for the MAT model (top right), which bears a resemblance to traffic lights, does not cause the model to remove correct detections.

responding digital-domain attacks, in which the attacker directly modifies the signal after it was received by the sensor and before it is processed by the model. The corresponding digital-domain

attack for the adversarial camera sticker is a type of universal adversarial perturbation (Moosavi-Dezfooli et al., 2017), while the digital adversarial patch attack (Brown et al., 2017; Anonymous, 2020) corresponds to physical patch attacks (Lee & Kolter, 2019; Huang et al., 2019).

We focus on increasing robustness against digital-domain attacks. Digital-domain attacks are strictly stronger than the corresponding physical-world attacks since they allow the attacker to have complete control over the change of the signal. In contrast, physical-world attacks need to be invariant under non-controllable effects such as scale, rotation, object position, and light conditions, which cannot be controlled by the attacker. Therefore, a system robust against digital-domain attacks is also robust against the corresponding physical-world attacks.

Currently, the most promising method for increasing robustness against adversarial attacks is adversarial training (Goodfellow et al., 2015; Madry et al., 2018). Adversarial training simulates an adversarial attack for every mini-batch and trains the model to become robust against such an attack. Adversarial training against digital-domain universal perturbations or patches is complicated by the fact that these attacks are computationally much more expensive than image-dependent adversarial attacks and existing approaches for speeding up adversarial training (Shafahi et al., 2019; Zhang et al., 2019; Zheng et al., 2020) are not directly applicable. Existing approaches for tailoring adversarial training to universal perturbations or patches either refrain from simulating attacks in every mini-batch (Moosavi-Dezfooli et al., 2017; Hayes & Danezis, 2018; Perolat et al., 2018), which bears the risk that the model easily overfits these fixed or rarely updated universal perturbations. Alternative approaches use proxy attacks that are computationally cheaper such as "universal adversarial training" (UAT) (Shafahi et al., 2018) and "shared adversarial training" (SAT) (Mummadi et al., 2019). These approaches face the challenge of balancing the implicit trade-off between simulating universal perturbation attacks accurately and keeping computation cost of the proxy attacks small.

We propose meta adversarial training (MAT)[1] , which falls into the category of proxy attacks. MAT combines adversarial training with meta-learning. We summarize the key novel contributions of MAT and refer to Section 3 for details:

- MAT amortizes the cost of computing universal perturbations by sharing information about optimal perturbations over consecutive steps of model training, which reduces the cost of generating strong approximations of universal perturbations considerably. In contrast to UAT (Shafahi et al., 2018), MAT uses meta-learning for sharing of information rather than joint training, which empirically generates stronger perturbations and a more robust model.
- MAT meta-learns a large set of perturbations concurrently. While a model easily overfits a single perturbation, even if it changes as in UAT, overfitting is much less likely for a larger set of perturbations such as those generated with MAT.
- MAT encourages diversity of the generated perturbations by assigning random but fixed target classes and step-sizes to each perturbation during meta-learning. This avoids that many perturbations focus on exploiting the same vulnerability of a model.

We perform an extensive empirical evaluation and ablation study of MAT on image classification and traffic-light detection tasks against a variety of attacks to show the robustness of MAT against universal patches and perturbations (see Section 4). We refer to Figure 1 for an illustration of MAT for universal patch attacks against traffic light detection.

## 2    RELATED WORK

We review work on generating universal perturbations, defending against them, and meta-learning.

**Generating Universal Perturbations**    Adversarial perturbations are changes to the input that are crafted with the intention of fooling a model's prediction on the input. Universal perturbations are a special case in which one perturbation needs to be effective on the majority of samples from the input distribution. Most work focuses on small additive perturbations that are bounded by some $\ell_p$-norm constraint. For example, Moosavi-Dezfooli et al. (2017) proposed the first approach by extending the DeepFool algorithm (Moosavi-Dezfooli et al., 2016). Similarly, Metzen et al. (2017) extended the iterative fast gradient sign method (Kurakin et al., 2017) for generating universal perturbations on semantic image segmentation. Mopuri et al. (2017; 2018) presented data-independent attacks and

---

[1]Code will be publicly released upon acceptance and can be found in the supplementary material.

Hayes & Danezis (2018) proposed using a generative model for learning a diverse distribution of universal perturbations. Li et al. (2019) presented a physical-world attack in which a translucent sticker is placed on the lens of a camera, which adds a universal perturbation to the image taken by the camera, and showed that this can fool an image classification system.

Other types of universal perturbations are so-called adversarial patches (Brown et al., 2017). In these universal patch attacks, the adversary can arbitrarily modify a small part of the image, typically a connected rectangular area, while leaving the remaining part of the image unchanged. Following Athalye et al. (2018), randomizing conditions such as location, rotation, scale, and lighting during the attack can make the universal patch sufficiently effective to fool the model when it is printed out and placed in the physical world. Later work has generalized these physical-world attacks to object detection (Lee & Kolter, 2019; Huang et al., 2019) and optical flow estimation (Ranjan et al., 2019).

**Defending Universal Perturbations**  First works for defending against universal perturbations are based on training a model against a fixed or slowly updated set/distribution of universal perturbations: Moosavi-Dezfooli et al. (2017) precompute a set of universal perturbations that are used during training, Hayes & Danezis (2018) learn a generative model of universal perturbations, and Perolat et al. (2018) build a slowly increasing set of universal perturbations concurrent to model training. A shortcoming of these approaches is that the model might overfit the fixed or slowly changing distribution of universal perturbations. However, re-computing universal perturbations in every mini-batch from scratch is prohibitively expensive. To address this issue, SAT Mummadi et al. (2019) trains a model against so-called shared perturbations. These shared perturbations do not have to be universal but only need to fool the model on a fixed subset of the batch. However, since the shared perturbations are recomputed in every mini-batch, it assumes a few gradient steps are sufficient to find strong perturbations from random initialization. In contrast, our method meta-learns strong initial perturbations. In UAT (Shafahi et al., 2018), training the neural network's weights and updating a single universal perturbation happen concurrently, which scales to a large dataset. However, our experiments in Section 4 indicate that a single incrementally and slowly updated perturbation is not sufficiently strong and diverse for making a model robust against all possible universal perturbations. Instead, our method meta-learns a large and diverse collection of perturbations during training.

For defending against adversarial patches, Chiang et al. (2020) proposed an approach of extending interval-bound propagation (Gowal et al., 2019) to the patch threat model. While this allows certification of robustness, it only scales to tiny patches and reduces clean accuracy considerably. Wu et al. (2020a) proposed the "defense against occlusion attack", which applies adversarial training to inputs perturbed with input-dependent adversarial patches placed at specific positions determined, for example, by the input gradient magnitude. Since they generate patches from scratch, they require an expensive optimization of the patch for every training batch. Moreover, robustness against stronger attacks such as those proposed in Section 4.1 remains unclear. Saha et al. (2019) hypothesize that vulnerability of object detectors against adversarial patches stems from contextual reasoning. Accordingly, they propose Grad-defense which penalizes strong dependence of object detections on their context in a data-driven manner, where dependence is determined by Grad-CAM (Selvaraju et al., 2019). Lastly, some non-adversarial data augmentation techniques resemble the universal adversarial patch scenario: they add a Gaussian noise patch (Lopes et al., 2019) or a patch from a different image (CutMix) (Yun et al., 2019) to each input. CutMix is conceptually very similar to the out-of-context defense (Saha et al., 2019). However, as demonstrated in our experiments in Section 4, even though these approaches increase robustness against occlusions, they are unlikely to increase robustness against universal patch attacks.

**Meta-Learning**  Gradient-based meta-learning methods such as MAML (Finn et al., 2017) or REPTILE (Nichol et al., 2018) allow learning initial parameters for a class of optimization tasks, so that one can find close-to-optimal parameters on a novel task from the distribution with a small number of gradient steps. Moreover, meta-learning can also be used to learn the task optimizer itself such as by Xiong & Hsieh (2020) in the context of adversarial training. While it is common to meta-learn initial weights for neural networks, we propose that these algorithms can also be used to meta-learn initial values for universal perturbations. In this work, we combine REPTILE with adversarial training because of the low computational overhead of REPTILE; however, in principle other gradient-based meta-learning methods could also be used as part of our method.

# 3 META ADVERSARIAL TRAINING

In this section, we propose a novel combination of adversarial training with meta-learning that trains models to be robust against universal perturbations.

## 3.1 PRELIMINARIES

Let $\mathcal{D}$ be a distribution over $d$-dimensional datapoints $x \in [0,1]^d$ and corresponding labels $y$, $\theta$ model parameters to be optimized, and $\mathcal{L}$ a loss function. Moreover, let $\mathcal{S}$ be the set of allowed perturbations and $\mathcal{F}$ be a function that applies a perturbation $\xi \in \mathcal{S}$ to a datapoint, potentially dependent on the label and some randomness $r \sim \mathcal{R}$. For universal perturbations (Moosavi-Dezfooli et al., 2017), one may choose $\mathcal{S} = \{\xi \mid ||\xi||_\infty \leq \epsilon\}$ for some small $\epsilon$ and $\mathcal{F}(x, \xi, r) = \text{Clip}_{[0,1]}[x + \xi]$. Alternatively, for universal patch attacks (Brown et al., 2017), we may define a mask $m \in \{0,1\}^d$, set $\mathcal{S} = [0,1]^{d_{patch}}$, and let $\mathcal{F}(x, \xi, r) = (1 - T(m, r)) \cdot x + T(m, r) \cdot T(\xi, r)$ with $T = T_l \circ \cdots \circ T_0$ being a sequence of stochastic transformations $T_i$ and some randomness $r \sim \mathcal{R}$ governing the stochasticity of the transformations. That is, each patch and mask are consistently transformed with $T$, e.g., translated, scaled, and rotated, and the transformed patch is applied to the input where the transformed mask is 1 and the input remains unchanged otherwise.

Following Mummadi et al. (2019), we define the universal adversarial risk as

$$\rho_{uni}(\theta) = \max_{\xi \in \mathcal{S}} \mathop{\mathbb{E}}_{(x,y) \sim \mathcal{D}, r \sim \mathcal{R}} \mathcal{L}(\theta, \mathcal{F}(x, \xi, r), y), \qquad (1)$$

where we drop the explicit dependence of $\rho_{uni}$ on $\mathcal{S}$, $\mathcal{D}$, and $\mathcal{R}$. Generally, we are interested in finding model parameters that minimize the universal adversarial risk, denoted as $\theta^* = \arg\min_\theta \rho_{uni}(\theta)$. This corresponds to the standard min-max saddle point formulation of adversarial training introduced by Madry et al. (2018), where we incrementally update the model parameters $\theta$ by computing $\theta_{t+1}$ based on $\nabla_{\theta_t} \rho_{uni}(\theta_t)$ (or more precisely an approximation of $\rho_{uni}(\theta_t)$). However, in contrast to standard adversarial training, the inner maximization problem is optimized over an expected value with respect to the data distribution $\mathcal{D}$ and potential randomness $\mathcal{R}$, making it more expensive to solve (even approximately). As the optimal $\xi_t$ of the inner maximization at step $t$ of the outer minimization depends on the parameter value $\theta_t$, this maximization of $\xi_t$ needs to be repeated in every step of the outer minimization, making the direct minimization of $\rho_{uni}(\theta)$ intractable.

Existing work has addressed this in different ways. One approach (Moosavi-Dezfooli et al., 2017; Hayes & Danezis, 2018; Perolat et al., 2018) relaxes the explicit dependence of $\xi_t$ on $\theta_t$ and computes a set or distribution over $\xi$ for some parameter checkpoints of $\theta$, and then applies these perturbations to the model while updating its parameters $\theta$. One shortcoming is that the outer minimization of $\theta$ can converge to a value for which the precomputed set or distribution over $\xi$ loses its effectiveness even though there are still some other choices of $\xi$ which are effective for $\theta$. Another approach is proposed by Mummadi et al. (2019): they instead replace the distribution $\mathcal{D}$ in the inner maximization with the current batch of the outer minimization. The effectiveness of this procedure hinges on the ability to efficiently approximate this inner maximization with few gradient steps. In summary, the main challenge of using adversarial training to increase model robustness against universal perturbations is efficiently approximating $\xi_t$ for the current $\theta_t$ in every step $t$ of the outer minimization.

## 3.2 META-LEARNING UNIVERSAL PERTURBATIONS

In contrast with the aforementioned approaches and similar to UAT (Shafahi et al., 2018), we exploit the property that one step of the outer minimization only applies a small change to $\theta$; thus, for consecutive steps $t$ and $t + 1$ of the outer minimization, the resulting inner maximization problems for finding $\xi_t$ and $\xi_{t+1}$ are closely related (Zheng et al., 2020). UAT exploits this property by initializing the inner maximization at $t + 1$ with the (approximate) solution for $\xi_t$ and performs a single gradient step on a single batch in the inner maximization at $t+1$. A potential shortcoming of this method is that it uses only a single gradient-step and thus implements joint training of parameters and perturbation, which does not allow capturing higher-order derivatives of the loss function (Nichol et al., 2018) and may therefore learn suboptimal initial parameters.

In order to address this shortcoming, we propose an approach to meta-learn initial values for universal perturbations – by approaching the optimization problems $\{\xi_{t_i}\}_{i=1}^N$ with gradient-based meta-learning:

in parallel to updating $\theta_t$ in the outer minimization, we meta-learn an initialization $\Xi_t$, which we refer to as the "meta-perturbation" at time step $t$ of the outer minimization. That allows for approximating the inner-optimization problem of $\xi_{t+1}$ with few gradient steps. More precisely, we use the REPTILE (Nichol et al., 2018) meta-learning algorithm with the iterative fast gradient sign method (I-FGSM) (Kurakin et al., 2017) task learner. In the inner maximization, we employ $K$ iterations of I-FGSM with the following update: $\xi_t^{(0)} = \Xi_t$ and $\xi_t^{(k+1)} = \Pi_{\mathcal{S}}\left[\xi_t^{(k)} + \alpha\,\mathrm{sgn}(\nabla_\xi \mathcal{L}(\theta_t, \mathcal{F}(x, \xi_t^{(k)}, r), y))\right]$, where $k$ indexes the inner maximization iterations, $\Pi_{\mathcal{S}}$ denotes projection on the set $\mathcal{S}$, and $\alpha$ the step size of I-FGSM. The key difference compared to standard I-FGSM and PGD (Madry et al., 2018) is that the initialization $\xi_t^{(0)}$ is neither constant nor randomly sampled but meta-learned.

The resulting perturbation $\xi_t^{(K)}$ is used two-fold: first, it is used with the REPTILE meta-learner for updating $\Xi$ with the following update: $\Xi_{t+1} = (1 - \sigma)\Xi_t + \sigma\xi_t^{(K)}$, where $\sigma$ is the learning rate of REPTILE. Second, $\xi_t^{(K)}$ is used in the next step of the outer minimization as an approximation of the optimal $\xi_t$ for the sample $(x, y)$ with randomness $r$. Learning the universal perturbation in UAT can be seen as a special case of our procedure for $K = 1$ and $\sigma = 1$.

We estimate the expected loss $\mathbb{E}_{(x,y)\sim\mathcal{D}, r\sim\mathcal{R}}\mathcal{L}(\cdot)$ in the I-FGSM task learner based on a single sample $(x, y) \sim \mathcal{D}, r \sim \mathcal{R}$. Moreover, we use the same sample in all $K$ steps of I-FGSM at time $t$ as well as in the outer minimization step of updating $\theta_t$. This provides us benefits of reduced variance and more efficient computation, however, at the cost of a biased estimate of $\rho_{uni}(\theta_t)$ – I-FGSM will converge to an $\xi_t^{(K)}$ that is overfit to $(x, y)$ and $r$. Compared to a perturbation optimized over the entire distributions $\mathcal{D}$ and $\mathcal{R}$, $\xi_t^{(K)}$ will incur a higher loss on the sample. Nevertheless, since we typically choose the number of I-FGSM steps $K \leq 10$, we expect only weak overfitting and the gains from reduced variance more than compensates for the increased bias.

## 3.3 META-LEARNING DIVERSE COLLECTIONS OF PERTURBATIONS

While the procedure proposed in Section 3.2 allows for meta-learning of a single meta-perturbation $\Xi$, one such meta-perturbation can easily get trapped in a local optimum, from which gradient-based meta-learning cannot easily escape. For instance, in a classification task with $C$ classes, there will likely be at least $C$ local optimal perturbations. More precisely, for each class, there is at least one optimum which corresponds to the perturbation that maximizes the model's prediction of this class. Hence, we propose a meta-learning approach that learns not just a single meta-perturbation but rather an entire set $\mathcal{P}$ of $P$ meta-perturbations, where the chosen $P$ should be proportional to $C$. For each sample, we select one of these meta-perturbations that will be used for initializing I-FGSM and later get updated by REPTILE.

However, meta-learning a set of meta-perturbations in this way with the same optimizer and objective will not automatically result in a diverse set of meta-perturbations. For instance, many of the perturbations might focus on exploiting similar weaknesses of a model such as triggering a misclassification to the same class. Moreover, utilizing the same task learner with the same step size might result in perturbations with similar properties. To alleviate this optimization problem, we encourage diversity of the generated set of meta-perturbations: for every meta-perturbation, we randomly assign a target and perform a targeted I-FGSM attack. This avoids many perturbations converging to similar patterns that fool the model into predicting the same target. Moreover, we also assign a randomly chosen fixed step size $\alpha$ for I-FGSM to every meta-perturbation. Larger step sizes correspond to meta-perturbations that explore the space of allowed perturbations more globally while smaller step sizes result in more fine-grained attacks. We empirically evaluate the effectiveness of these heuristics in Section 4.

## 3.4 META ADVERSARIAL TRAINING

We summarize the proposed meta adversarial training (MAT) in Algorithm 1. The function $\mathrm{INIT}^P$ (see Algorithm 2 in the appendix) is responsible for initializing $\mathcal{P}$ consisting of $P$ meta-perturbations $\Xi_i$ along with corresponding targets $y_i^{target}$ and step-sizes $\alpha_i$. For classification tasks, we select the target as one of the classes in a round-robin fashion. Moreover, we select the step size log-uniformly from $[0.0001, 0.1]$. We initialize the meta-perturbations by either sampling uniform randomly from $[0, 1]^d$ or by (sub-sampling) an actual data-point, which corresponds to an on-manifold initialization

---

**Algorithm 1** Meta Adversarial Training

---

1: **Input:** data $\mathcal{D}$, initial parameters $\theta$, application fct. $\mathcal{F}$, loss-fct. $\mathcal{L}$, REPTILE learn-rate $\sigma$
2: $\mathcal{P} \leftarrow \text{INIT}^P(\mathcal{D},\text{"data"})$ # Initialize $P$ tuples of meta-perturbation, target, and step size
3: # $t_{max}$ steps of outer minimization, we drop subscript $t$ below for readability
4: **for** $t$ **in** $\{0, \ldots, t_{max} - 1\}$ **do**
5: $\quad (x, y) \sim \mathcal{D}$ # Sample datapoint
6: $\quad$ # Select meta-pert. $\Xi$ and corresponding target $y^{target}$, step-size $\alpha$, and randomness $r$ from $\mathcal{P}$
7: $\quad \Xi, y^{target}, \alpha, r \leftarrow \text{SELECT}^F(\mathcal{P}, x, y, \theta_t, \mathcal{F}, \mathcal{L}, \mathcal{R})$
8: $\quad \xi \leftarrow \text{I-FGSM}^K(\Xi, x, y^{target}, \theta_t, \mathcal{F}, \alpha, r)$ # Inner maximization initialized with meta-pert. $\Xi$
9: $\quad x^{pert} \leftarrow \mathcal{F}(x, \xi, r)$ # Apply perturbation $\xi$ to $x$
10: $\quad \theta \leftarrow \text{OPT}(\mathcal{L}, \theta, x^{pert}, y)$ # Outer minimization step with optimizer OPT, for instance SGD
11: $\quad \Xi \leftarrow (1 - \sigma)\Xi + \sigma\xi$ # Meta-learning update of meta-perturbation $\Xi$ with REPTILE
12: $\quad \mathcal{P} \leftarrow \text{UPDATE}(\mathcal{P}, \Xi)$ # Replace updated meta-perturbation in $\mathcal{P}$
13: **end for**

---

akin to CutMix (Yun et al., 2019). This data-initialization was concurrently proposed by Yang et al. (2020b), and Yang et al. (2020a) found that such texture patches can be adversarial even without further optimization. The function $\text{SELECT}^F$ (see Algorithm 3 in the appendix) uniform randomly samples $F$ trials of $(\Xi, y^{target}, \alpha)$ from $\mathcal{P}$ with randomness $r \sim \mathcal{R}$ and returns the trial which maximizes the loss $\mathcal{L}(\theta, \mathcal{F}(x, \Xi, r), y)$.

Line 8-11 present the core of MAT consisting of (i) inner maximization of a perturbation $\xi$ that was initialized from a meta-perturbation $\Xi$ with I-FGSM$^K$ (Line 8), (ii) a step of outer minimization of $\theta_t$ with an optimizer like SGD on a pair of perturbed input $x^{pert}$ and corresponding label $y$ (Line 10), and (iii) the meta-learning update of the respective meta-perturbation $\Xi$ with REPTILE (Line 11). While Algorithm 1 shows the procedure for a batch size equal to one, we can easily run it also for larger batch sizes. The only required change is that REPTILE-based meta-learning can deal with the situation where the same meta-perturbation is selected and optimized for several elements in a batch. In this case, the meta-learning update becomes $\Xi = (1 - \sigma)\Xi + \sigma\frac{1}{N}\sum_{i=1}^{N}\xi_i$ for $N$ perturbations $\xi_i$ that were initialized with the same meta-perturbation $\Xi$.

We briefly summarize the main advantages of MAT compared to prior work: as opposed to UAT, MAT meta-learns a diverse set of meta-perturbations with I-FGSM$^K$ concurrently to model training rather than jointly training model parameters and a single perturbation with FGSM. Compared to SAT (Mummadi et al., 2019), MAT does not treat every inner maximization problem independently but meta-learns strong initializers, allowing MAT to find stronger perturbations with no more computational cost than standard adversarial training (see Section A). In contrast to the work of Moosavi-Dezfooli et al. (2017); Hayes & Danezis (2018); Perolat et al. (2018), MAT computes novel perturbations in every iteration of model training (outer minimization). We would also like to note that MAT meta-learns perturbations but not model weights and thus results in a standard trained model that does not require test-time adaptation.

## 4 EXPERIMENTS

We evaluate the performance of MAT, ablated versions of MAT, and baselines. We present results for universal patch attacks on image classification on Tiny ImageNet (Tin) (results for universal perturbations attacks are provided in Section C.1.2). In addition, we present results for a universal patch attack for an object detection task on the Bosch Small Traffic Lights Dataset (Behrendt & Novak, 2017). For a detailed description of the implementation of the experiments, we refer to Section B in the appendix. In order to compare different methods for increasing robustness against universal patches, a reliable way of evaluating their robustness is required, which we outline next.

### 4.1 RELIABLE ROBUSTNESS EVALUATION

We outline strong attacks for reliably evaluating the robustness of trained models against universal perturbations and patches. Importantly, we do not use the meta-learned perturbations $\Xi_i$ as this might result in a biased robustness evaluation. Instead, we extend PGD (Madry et al., 2018) in a similar

way as Mummadi et al. (2019) by rewriting $\rho_{uni}$ from Equation (1) to $\rho_{uni}(\theta) = \max_{\xi \in \mathcal{S}} \rho(\theta, \xi)$ with

$$\rho(\theta, \xi) = \mathop{\mathbb{E}}_{(x,y) \sim \mathcal{D}, r \sim \mathcal{R}} \mathcal{L}(\theta, \mathcal{F}(x, \xi, r), y),$$

and then use the estimate

$$\hat{\rho}(\theta, \xi) = \frac{1}{N} \sum_{i=1}^{N} \mathcal{L}(\theta, \mathcal{F}(x_i, \xi, r_i), y_i)$$

based on samples $(x_i, y_i) \sim \mathcal{D}$ and $r_i \sim \mathcal{R}$. Finally, we define stochastic projected gradient descent (S-PGD) as $\xi^{(0)} \sim \mathcal{S}$ and $\xi^{(k)} = \Pi_{\mathcal{S}} \left[ \xi^{(k-1)} + \alpha \operatorname{sgn}(\nabla_\xi \hat{\rho}(\theta, \xi^{(k-1)})) \right]$. Note that S-PGD uses different $x_i, y_i, r_i$ in every step when estimating $\hat{\rho}$.

In general, S-PGD will converge to local optima; namely, $\xi^{(K)}$ obtained after K steps of S-PGD will not necessarily be the global maximizer of $\hat{\rho}(\theta, \xi)$. To account for this, we propose three extensions of S-PGD: Firstly, since the initialization of $\xi_0$ will generally affect the quality of $\xi^{(K)}$, we propose an alternative initialization akin to CutMix (Yun et al., 2019) where we initialize $\xi^{(0)}$ based on a datapoint $x \sim \mathcal{D}$. For universal patch attacks, we downsample or crop $x$ to the patch size, whereas for universal perturbation attacks, we scale its intensity range such that $x \in \mathcal{S}$. This initialization becomes even more effective if we sample many $x \sim \mathcal{D}$ and select the one for initializing $\xi^{(0)}$ which would maximize $\hat{\rho}(\theta, \xi^{(0)})$. We denote this initialization as *data initialization*.

Secondly, we take inspiration from recently proposed *low-frequency attacks* (Guo et al., 2019; Sharma et al., 2019): we modify the process of adding a perturbation $\xi$ to an input to $\mathcal{F}(x_i, LP(\xi, u), r_i)$, where $LP(\xi, u)$ denotes a low-pass filter with cutoff-frequency $u$. To achieve this, we follow Jo & Bengio (2017) and create a centered radial mask with radius $u$. The patch is transformed into frequency space and multiplied by the radial mask. The result is transformed back to image space and thus yields the patch to be applied to the image. While this makes the attack weaker in principle since only low-frequency perturbations are possible, we observe that in practice, it can lead to a more well-behaved optimization problem and result in S-PGD converging to stronger perturbations.

Thirdly, we perform a *transfer attack*, in which we run an attack after every epoch of model training. We initialize $\xi^{(0)}$ with one of the $\xi^{(K)}$ found in previous epochs, namely the one that would maximize $\hat{\rho}(\theta, \xi^{(0)})$. After every 5 epochs, we run an additional S-PGD attack from randomly initialized $\xi^{(0)}$. This transfer attack helps identify cases where universal perturbations found in early epochs remain effective against the model but in later epochs are no longer found when running S-PGD attacks from random or data initialization.

## 4.2 IMAGE CLASSIFICATION ON TINY IMAGENET

We evaluate robustness against universal patches of size 24x24 pixel that cover approximately 14% of the image. Patches are randomly translated from the center of the image by at most 26 pixels.

We train every model for 75 epochs with SGD, an initial learning rate of 0.033, a cosine decay learning rate scheduler, momentum 0.9, and a batch size of 128. We use a ResNet (He et al., 2016), train it from scratch, and follow Xie & Yuille (2020) by replacing batch normalization with group normalization (Wu & He, 2019) and weight standardization (Qiao et al., 2019). We use $K = 5$ iterations of I-FGSM in AT (Madry et al., 2018), SAT (Mummadi et al., 2019), and MAT. For UAT (Shafahi et al., 2018), we use $K = 1$ following their recommendation. For SAT, we use sharedness 128. We note that all adversarial training baselines were trained against patch attacks.

For every setting, we perform 5 independent runs. We evaluate the robustness against 2500-step-S-PGD with a batch size of 64 and random initialization, data initialization (data samples resized to patch size), and low-frequency filter, and the transfer attack (see Section 4.1). For the S-PGD settings, we perform a grid search (see Table 4) over step sizes $\alpha \in \{0.0001, 0.00033, 0.001, 0.0033, 0.01, 0.033, 0.1\}$ and momentum $\gamma \in \{0, 0.9, 0.99\}$ independently for every trained model and report the minimal accuracy. Finally, we report the minimal accuracy across all attacks.

Results are summarized in Table 1 (more details can be found in Section C.1). We observe that a model trained with standard empirical risk minimization offers no robustness against any of the

Table 1: Accuracy (mean over 5 runs) of different methods on Tiny ImageNet on clean data (CL) and against universal patch attacks with random init (RI), init with a cropped image patch (DI), low-frequency filter (LF), transfer attacks (Tr), and worst across all four attacks (Min).

| SETTING | | CL | RI | DI | LF | TR | MIN |
|---|---|---|---|---|---|---|---|
| STANDARD | | 0.55 | 0.03 | 0.03 | 0.07 | 0.03 | 0.02 |
| CUTMIX (YUN ET AL., 2019) | | 0.57 | 0.03 | 0.03 | 0.07 | 0.03 | 0.02 |
| PATCHUNIFORM (LOPES ET AL., 2019) | | 0.57 | 0.04 | 0.06 | 0.10 | 0.04 | 0.03 |
| AT (MADRY ET AL., 2018) | | 0.57 | 0.06 | 0.07 | 0.12 | 0.17 | 0.06 |
| SAT (MUMMADI ET AL., 2019) | | 0.58 | 0.07 | 0.07 | 0.12 | 0.33 | 0.06 |
| UAT (SHAFAHI ET AL., 2018) | | 0.49 | 0.41 | 0.09 | 0.15 | 0.12 | 0.09 |
| MAT | (FULL) | **0.59** | 0.56 | 0.55 | 0.53 | **0.55** | **0.53** |
| | (RANDOM INIT) | 0.58 | 0.56 | 0.32 | 0.26 | 0.52 | 0.23 |
| | (UNTARGETED) | 0.58 | 0.56 | **0.56** | **0.56** | 0.50 | 0.50 |
| | ($F = 1$) | 0.58 | 0.53 | 0.53 | 0.50 | 0.52 | 0.48 |
| | ($K = 1$) | 0.58 | 0.55 | 0.53 | 0.53 | 0.46 | 0.45 |
| | ($\sigma = 1.0$) | 0.58 | **0.57** | 0.55 | 0.55 | 0.50 | 0.50 |

evaluated attacks. Similarly, PatchUniform (akin to Lopes et al. (2019) but with uniform rather than GaussianNoise) or CutMix (Yun et al., 2019) augmentation do not offer robustness against strong patch attacks. When comparing the baseline adversarial defenses (Madry et al., 2018; Mummadi et al., 2019; Shafahi et al., 2018), none of them exhibit more than trivial robustness. UAT provides relatively high robustness against S-PGD with random initialization, even though we perform an extensive grid search over hyperparameters of S-PGD. However, this robustness does not carry over to other attack variants. Closer inspection of AT and SAT (see Figure 4 in the appendix) shows that also for these methods only very specific combinations of learning rate and momentum allow for discovery of effective patches from random initialization. In summary, since S-PGD with random initialization, a single step-size, and momentum value is often used as the default evaluation, we suspect that some prior work might have overestimated the robustness of existing methods.

In contrast, MAT (full) with standard parameters (INIT with data initialization and $P = 1000$ meta-perturbations, targeted attacks, $F = 5$ in SELECT, $K = 5$ iterations in I-FGSM, REPTILE learning rate $\sigma = 0.25$) shows high robustness against all attack variants. When ablating MAT, choosing a random initialization in INIT is most problematic – it results in similar but less severe overfitting to randomly initialized attacks as UAT. Also, ablating towards joint training ($\sigma = 1.0$ and $K = 1$) deteriorates performance relative to meta-learning in MAT (full). In addition, enforcing diversity in meta-perturbations via targeted attacks in MAT (full) is responsible for a small increase in robustness compared to untargeted learning of meta-perturbations. Finally, taking the worst over $F = 5$ samples in SELECT outperforms random sampling ($F = 1$).

We observed that baseline adversarial training procedures like AT, SAT, and UAT are very sensitive to the step size $\alpha$ used in the inner maximization. Although we have tuned their step sizes to some extent, there might be some effective $\alpha$ with which their performance would improve. However, MAT does not require us to tune $\alpha$ at all since it uses different learning rates for every meta-perturbation, and thus it covers a broad range of step sizes automatically. Most importantly, MAT offers increased robustness without affecting clean performance. In contrast, MAT acts as an effective regularizer and reduces overfitting compared to standard training and achieves the strongest clean performance among all methods, surpassing standard training by 4 percentage points. Even against the strongest patches, MAT only loses 2 percentage points accuracy relative to standard training on clean data, despite the relatively large patch size. We provide illustrations of patches in Section D and show in Section A that MAT has similar computational cost as standard adversarial training.

## 4.3 OBJECT DETECTION ON BOSCH SMALL TRAFFIC LIGHTS DATASET

We evaluate robustness of a traffic light detector based on YoloV3 (Redmon & Farhadi, 2018). This attack scenario is a good proxy for physical-world attacks on automated driving systems since traffic light detection crucially relies on camera-based perception. We add 64x64 patches to 1280x704 images, covering 0.45% of the image, and add random translations from the center by up to $(512, 282)$

Table 2: Mean recall (left) and mean average precision (right) of different methods on clean data (CL) and against universal patch attacks with random init (RI), init with a cropped image patch (DI) and low-frequency filter (LF) on Bosch Small Traffic Lights (Behrendt & Novak, 2017).

| Recall | CL | RI | DI | LF | MIN | mAP | CL | RI | DI | LF | MIN |
|---|---|---|---|---|---|---|---|---|---|---|---|
| STANDARD | **0.47** | 0.37 | 0.38 | 0.43 | 0.37 | | **0.41** | 0.09 | 0.10 | 0.16 | 0.09 |
| UAT | **0.47** | **0.45** | **0.45** | **0.45** | **0.45** | | **0.41** | **0.40** | 0.29 | 0.26 | 0.26 |
| MAT (DEFAULT) | 0.45 | 0.43 | 0.43 | 0.43 | 0.43 | | **0.41** | 0.39 | 0.35 | 0.35 | 0.35 |
| (+DATA INIT) | 0.45 | 0.43 | 0.43 | 0.42 | 0.42 | | **0.41** | 0.38 | **0.39** | **0.39** | **0.38** |
| ($K = 1$) | 0.46 | 0.44 | 0.43 | 0.44 | 0.43 | | **0.41** | **0.40** | 0.31 | 0.27 | 0.27 |
| ($K = 1, P = 1$) | 0.46 | 0.44 | 0.43 | 0.43 | 0.43 | | **0.41** | **0.40** | 0.21 | 0.17 | 0.17 |

pixels. We train the models with ADAM for 15 epochs with batch size 1 and learning rate 0.0001. We replace batch normalization by group normalization with weight standardization.

We evaluate the effectiveness of universal patch attacks using two metrics: mean Average Precision (mAP) and mean recall over classes for a fixed confidence threshold. While mAP captures both non-existent detections caused by the patch (false positives) and correct detections missed by the model (false negatives), mean recall focuses only on the latter. In other words, recall captures "blindness" attacks (Saha et al., 2019) that could be more dangerous in real life scenarios. We set the confidence threshold to 0.3, the non-maximum suppression threshold to 0.1, and the IOU threshold for evaluating true positives to 0.1, respectively.

For attacks with S-PGD, we run 4000 steps with a batch size of 4, fix momentum to $\gamma = 0.9$, and perform a grid search (see Table 6) over step sizes $\alpha \in \{0.1, 0.01, 0.001, 0.0001\}$ and cutoff frequency $u \in \{25, 50, 100\}$ of the optional low-pass filter. Moreover, we also conduct a grid search over three options for the loss maximized by the attacker: the standard loss also used for model training, the standard loss subtracting the objectness loss as proposed by Saha et al. (2019), and the standard loss ignoring all false positives. The last loss variant is also well suited for "blindness" attacks since it accentuates false negatives. For MAT (default), we use random initialization in INIT, $P = 10$ meta-perturbations, $F = 1$ in SELECT, $K = 3$ iterations in I-FGSM, and learning rate $\sigma = 0.25$ in REPTILE as the default configuration.

Table 2 summarizes the results (more details can be found in Section C.2). In general, there are no systematic differences between training methods on clean data. Standard training faces a drop in mean recall when a universal patch is added. Thus, standard training is likely to be susceptible to physical-world blindness attacks, which could for example cause the model to ignore real traffic lights. In contrast, UAT and all variants of MAT are very robust against the tested blindness attacks even in the digital domain. In terms of the mAP, UAT faces a considerable drop for patch attacks initialized with data crops and with a low-frequency filter. A similar but weaker effect can also be observed for MAT with the default configuration. Both methods therefore detect non-existent traffic lights on the patch or in its vicinity. Interestingly, false positive detections often resemble traffic lights (see Figure 1 and Section E). Despite this resemblance, a human would not be fooled by these patches. The same holds true for MAT with data initialization in INIT: its high mAP indicates that it is very robust in terms of prevention of additional false positives. When ablating MAT, we observe that its mAP deteriorates as the configuration approaches UAT ($K = 1$ and $P = 1$). We conclude that all aspects of MAT are essential for achieving maximal robustness.

## 5 CONCLUSION

We propose meta adversarial training (MAT), a novel combination of adversarial training with meta-learning that allows the increase of model robustness against universal perturbations and patches with little computational overhead. Moreover, we show that prior work, which was assumed to be robust, can be fooled by stronger attacks. In contrast, MAT remains robust against all evaluated attacks. Our results show that further research into attacks or alternatively scaling up certification procedures (Chiang et al., 2020) is required for reliably evaluating robustness against universal perturbations. On the other hand, our results also indicate that physical-world attacks will become considerably more difficult against models trained with MAT.

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

# META ADVERSARIAL TRAINING - SUPPLEMENTARY MATERIAL

## A COMPUTATIONAL COST OF META ADVERSARIAL TRAINING

Computational cost of MAT is dominated by the number of forward passes $n_{fp}$ and backward passes $n_{bp}$ through the network for a single iteration of model training (the outer loop). Adversarial training (AT) with $K$-step PGD incurs $(K + 1) * (n_{fp} + n_{bp})$ cost for one iteration: $K * (n_{fp} + n_{bp})$ for generating the perturbation and $1 * (n_{fp} + n_{bp})$ for one step of model training. Similarly, the cost of MAT (with REPTILE as in our experiments) for inner maximization and one step of outer minimization is $(K + 1) * (n_{fp} + n_{bp})$. Additionally, selecting a meta-perturbation from $F$ samples in Algorithm 3 incurs a cost of $F * n_{fp}$ for $F > 1$. The additional cost is 0 for $F = 1$ because the meta-perturbation is sampled randomly in this case and its loss need not be computed. The cost of REPTILE itself is negligible, because it is a simple convex combination. Therefore, the cost of MAT ($F = 1$) and that of AT are comparable and MAT ($F = 1$) clearly outperforms AT in Table 1. Also for a small $F$ such as $F = 5$ for MAT (full) in Table 1, MAT+REPTILE does not incur considerably higher cost than AT for the same $K$. The key point is that due to better initialization from the set of meta-perturbations, a small $K$ can be chosen for MAT, whereas $K$ would need to be very large for PGD in order to create equally strong attacks.

## B IMPLEMENTATION DETAILS

### B.1 SUBPROCEDURE INIT$^P$

We present details on subprocedure INIT$^P$ in Algorithm 2. Two relevant parameters of INIT$^P$ are described below.

**Initialization of Meta-Perturbations** As described in Section 3.2, meta adversarial training (MAT) meta-learns a set $\mathcal{P}$ of $P$ meta-perturbations $\Xi_i$, where $i$ indexes $P$. Similar to the attack initialization described in Section 4.1, these meta-perturbations can be initialized in INIT$^P$ in two ways as follows:

- *Random initialization*: sampling randomly from a uniform distribution over the space of allowed perturbations $\mathcal{S}$.
- *Data initialization*: this initialization sub-samples actual data points from the training dataset and corresponds to an on-manifold initialization that follows the data distribution. To generate universal patches, we downsample or crop the data points. To create universal perturbations, we scale the intensity of the data points to the range of $\mathcal{S}$.

**Number of Meta-Perturbations** The number of meta-perturbations $P$ is chosen roughly proportional to the number of classes of the dataset regardless of classification or object detection tasks. We choose $P = 1000$ for Tiny ImageNet, which has 200 classes. For Bosch Small Traffic Lights Dataset, we choose $P = 10$ because the dataset only has 4 classes.

---

**Algorithm 2** INIT$^P$

---
1: **Input:** number of meta-perturb. $P$, data $\mathcal{D}$, initialization type "init"
2: $\mathcal{P} \leftarrow \{\}$
3: **for** $i$ **in** $\{1, \ldots, P\}$ **do**
4:     $y^{target} \leftarrow i \bmod C$ # round-robin target class, $C$ being the number of classes
5:     **if** init = "random" **then**
6:         $\Xi \sim \text{UNIFORM}([0, 1]^{d_{patch}})$
7:     **else if** init = "data" **then**
8:         $x \sim \text{UNIFORM}(\mathcal{D}|_{y=y^{target}})$ # Select datapoint labeled with target class uniformly
9:         $\Xi \leftarrow \text{RESIZE}(x, d_{patch})$ # Resize datapoint to appropriate dimensionality
10:     **end if**
11:     $\alpha \sim \text{LOGUNIFORM}(0.0001, 0.1)$
12:     $\mathcal{P} \leftarrow \mathcal{P} \cup \{(\Xi, y^{target}, \alpha)\}$
13: **end for**
14: Return $\mathcal{P}$

---

---

**Algorithm 3** SELECT$^F$

---

1: **Input:** number of trials $F$, set of meta-perturb. $\mathcal{P}$, input $x$, label $y$, parameters $\theta$, application fct. $\mathcal{F}$, loss-fct. $\mathcal{L}$, random generator $\mathcal{R}$
2: $\Xi_{opt}, y_{opt}^{target}, \alpha_{opt} \sim \text{UNIFORM}(\mathcal{P})$ # Sample uniform randomly from $\mathcal{P}$
3: $r_{opt} \sim \mathcal{R}$
4: **if** $F = 1$ **then**
5:     Return $(\Xi_{opt}, y_{opt}^{target}, \alpha_{opt}, r_{opt})$
6: **end if**
7: $l_{opt} \leftarrow \mathcal{L}(\theta, (\mathcal{F}(x, \Xi_{opt}, r_{opt}), y))$ # loss on perturbed data, according label for parameters $\theta$
8: **for** $i$ **in** $\{1, \ldots, F\}$ **do**
9:     # Find worst (in terms of loss) out of $F$ trials
10:     $\Xi, y^{target}, \alpha \sim \text{UNIFORM}(\mathcal{P})$
11:     $r \sim \mathcal{R}$
12:     $l \leftarrow \mathcal{L}(\theta, (\mathcal{F}(x, \Xi, r), y))$
13:     **if** $l > l_{opt}$ **then**
14:         $(\Xi_{opt}, y_{opt}^{target}, \alpha_{opt}, r_{opt}) \leftarrow (\Xi, y^{target}, \alpha, r)$
15:         $l_{opt} \leftarrow l$
16:     **end if**
17: **end for**
18: Return $(\Xi_{opt}, y_{opt}^{target}, \alpha_{opt}, r_{opt})$

---

### B.2 SUBPROCEDURE SELECT$^F$

We present details on the sub-procedure SELECT$^F$ in Algorithm 3. We note that the special choice $F = 1$ corresponds to a uniform random sampling of a meta-perturbation (and corresponding target class, step-size, and randomness). For $F > 1$, SELECT$^F$ requires $F$ additional evaluations of the loss functions (and thus forward passes through the model) since the sample with the maximal loss is selected.

### B.3 IMAGE CLASSIFICATION ON TINY IMAGENET

To evaluate the robustness of a model trained with MAT against universal patch attacks and universal perturbation attacks, we compare its performance with other training approaches such as Standard, CutMix (Yun et al., 2019), PatchUniform, adversarial training (AT) (Madry et al., 2018), shared adversarial training (SAT) (Mummadi et al., 2019), and universal adversarial training (UAT) (Shafahi et al., 2018). Please note that this evaluation is an ablation study of MAT, namely, we configure MAT in a way that is similar to each training approach. Detailed configurations are shown for universal patches in Table 3 and for universal perturbations in Table 7.

**Model Architecture** For each setting, we train a ResNet V1 model (He et al., 2016) from scratch on the Tiny ImageNet dataset (Tin). This ResNet model contains 4 residual stacks, where each stack consists of 2 residual blocks. The stacks have 64, 128, 256, and 512 channels and spatial resolution of 64x64, 32x32, 16x16, 8x8, respectively. We employ ReLU as its activation function. Each convolutional layer has a stride of 1, kernel size of 3, group normalization with weight standardization, SAME padding, "he_normal" kernel initialization, and a weight decay of $1 \cdot 10^{-4}$ on the kernel weights.

**Model Training** Each model is trained with 24x24 pixel patches applied to the 64x64 pixel input images, namely, a patch covers approximately 14% of the image. Patches are randomly translated from the center of the image by up to 26 pixels during training. We train each model with SGD for 75 epochs, an initial learning rate of 0.033, a cosine decay learning rate scheduler, momentum of 0.9, and a batch size of 128. For each setting, we perform 5 independent runs with 5 different seeds. Details regarding adversarial training procedures are shown in the Table 3. The most crucial parameter for AT, SAT, and UAT is the step size $\alpha$ of I-FGSM. For AT, we follow MAT and sample the step size per datapoint randomly from a log-uniform distribution over $[0.0001, 0.1]$. Since UAT and SAT only update a single patch per batch, this random sampling strategy is not feasible on a per-batch level. Instead, we use a fixed $\alpha$; more specifically, we use 0.2 for SAT such that I-FGSM can reach any value in $[0, 1]^d$ in K=5 iterations. Since UAT updates the perturbation iteratively over

| SETTING | CUTMIX | PATCHUNIFORM | AT | SAT | UAT | MAT (FULL) |
|---|---|---|---|---|---|---|
| PATCH INITIALIZATION | DATA | RANDOM | RANDOM | RANDOM | RANDOM | DATA |
| WORST OVER $F$ SAMPLES | – | – | 1 | 1 | 1 | 5 |
| REPTILE LEARNING RATE $\sigma$ | – | – | 0 | 0 | 1 | 0.25 |
| NUMBER OF META-PATCHES $P$ | – | – | – | – | 1 | 1000 |
| I-FGSM STEP-SIZE $\alpha$ | – | – | [0.0001, 0.1] | 0.2 | 0.01 | [0.0001, 0.1] |
| I-FGSM ITERATIONS $K$ | – | – | 5 | 5 | 1 | 5 |
| SHAREDNESS | – | – | – | 128 | – | – |

Table 3: Configuration of training procedures against universal adversarial patch attacks for image classification. We denote irrelevant entries as '−'.

| Parameter | Values |
|---|---|
| Patch Initialization | random, data |
| Step Size $\alpha$ | 0.0001, 0.00033, 0.001, 0.0033, 0.01, 0.033, 0.1 |
| Momentum $\gamma$ | 0, 0.9, 0.99 |
| Cutoff Frequency | off, 12 |
| Number of iteration (S-PGD) | 2500 |
| Total Step Size Decay | 0.01 |
| Batch Size | 64 |

Table 4: Configuration grid of attacks against the classification tasks.

the batches, a smaller value for $\alpha$ is feasible here and we employ $\alpha = 0.01$. We did not tune these choices for $\alpha$ extensively but note that MAT does not require any tuning of the step size $\alpha$.

**Model Evaluation**   As described in Section 4.1, we propose strong attacks for reliably evaluating the robustness of trained models against universal patch attacks and universal perturbation attacks by optimizing the perturbations using S-PGD. We propose two initialization methods for S-PGD. Additionally, we utilize the low-frequency attack described above. The S-PGD step size $\alpha$ is exponentially decayed with a total decay of $0.01$. For evaluation of each model's robustness, we perform S-PGD attacks over the parameter grid given in Table 4. The attack results can be found in Subsection C.1.

### B.4   OBJECT DETECTION ON BOSCH SMALL TRAFFIC LIGHTS DATASET

We describe the experimental details for training robust traffic light detectors against universal patch attacks.

**Model Training**   For each training procedure, we train a Yolo V3 model (Redmon & Farhadi, 2018) from scratch on Bosch Small Traffic Lights Dataset (Behrendt & Novak, 2017). The model has three network outputs on each scale as implemented in the original paper. For each DarkNet conv layer, we replace batch normalization with group normalization and use weight standardization. To interpret the network outputs of Yolo V3, we set the confidence threshold to 0.3. This means only the predictions with an objectness score $> 0.3$ count as valid predictions. The non-maximum suppression threshold is set to 0.1, that means we prune the predictions when their bounding boxes overlap with IoU $> 0.1$.

Each model is trained with 64x64 pixel patches applied to the input images resized to 1280x704 – both width and height of the resized images are a multiple of 32 because a grid cell's size is 32x32; thus, a patch covers 0.45% of the image. Patches are randomly translated from the center of the image by up to (512, 282) pixels during training. We ensure that translated patches do not overlap with any ground-truth traffic-light annotation. We train the model with ADAM for 15 epochs, an initial learning rate of 0.0001, a cosine decay learning rate scheduler, and a batch size of 1. We compare the accuracy against universal patch attacks of UAT and MAT variants. The configuration details are shown in Table 5.

**Model Evaluation**   As described in Section 4.3, in order to evaluate the effectiveness of the universal patches as well as the robustness of the model, we apply two metrics to the evaluation procedure -

| SETTING | UAT | MAT (DEFAULT) | MAT(+DATA) | MAT ($K$=1) | MAT($K$=1,$P$=1) |
|---|---|---|---|---|---|
| PATCH INITIALIZATION | RAND | RAND | DATA | RAND | RAND |
| WORST OVER $F$ SAMPLES | 1 | 1 | 1 | 1 | 1 |
| REPTILE LEARNING RATE $\sigma$ | 1.0 | 0.25 | 0.25 | 0.25 | 0.25 |
| NUMBER OF PATCHES $P$ | 1 | 10 | 10 | 10 | 1 |
| I-FGSM STEP-SIZE $\alpha$ | 0.01 | $[0.0001, 0.1]$ | $[0.0001, 0.1]$ | $[0.0001, 0.1]$ | $[0.0001, 0.1]$ |
| I-FGSM ITERATIONS $K$ | 1 | 3 | 3 | 1 | 1 |

Table 5: Configuration of MAT with different approaches against universal adversarial patch attacks for object detection.

| Parameter | Values |
|---|---|
| Patch Initialization | random (RI), data crop (DI) |
| Number of Steps (S-PGD) | 4000 |
| Batch Size | 4 |
| Step Size $\alpha$ | 0.1, 0.01, 0.001, 0.0001 |
| Total Step Size Decay | 0.01 |
| Momentum $\gamma$ | 0.9 |
| Cutoff Frequency $u$ | 25, 50, 100, 250 |
| Loss | standard, no objectness loss, ignoring false positives |

Table 6: Configuration grid of attacks against the detection tasks.

mean Average Precision (mAP) and mean recall over classes with the IoU threshold of 0.1, which determines true positives between predicted bounding boxes and the ground truth. For generating universal patches, we use S-PGD with 4000 steps with a batch size of 4. To find the strongest patches, we perform a grid search over step sizes $\alpha \in \{0.1, 0.01, 0.001, 0.0001\}$, a fixed momentum $\gamma$ of 0.9, and a cutoff frequency $u \in \{25, 50, 100, 250\}$ of an optional low-pass filter. In addition, we also conduct a grid search over three different options for the loss that is maximized by the attacker - 1) the standard loss that is also used during training, 2) the standard loss subtracting the objectness loss, and 3) the standard loss ignoring all false positives. Similar to the previous initialization approaches in Section B.3, the perturbations for these attacks are initialized in two different ways - randomly or from a cropped image of Bosch Small Traffic Lights Dataset. Each configuration is a unique combination of an initialization, a step size, a cutoff frequency, and a loss found through the grid search. The parameter grid is summarized in Table 6. More result details can be found in Section C.2.

## C  RESULT DETAILS

### C.1  IMAGE CLASSIFICATION ON TINY IMAGENET

#### C.1.1  ROBUSTNESS AGAINST UNIVERSAL PATCH ATTACKS

In Figure 2, we compare learning curves of MAT models against the transfer attack (see Section 4.1) between different settings during training. The left plot shows that initializing the meta-perturbations via *data initialization* leads to higher universal adversarial accuracy compared to *random initialization*. The middle plot shows that the model trained with targeted meta-perturbations is more robust than the model trained with untargeted meta-perturbations, because targeted meta-perturbations allow for a greater diversity. The right plot shows results of randomly choosing a patch ($F = 1$), selecting the worst patch from $F = 5$ samples, and selecting the worst patch from $F = 10$ samples. Training the model with more than one sample ($F > 1$) improves the model's robustness but robustness saturates for $F = 10$ while larger $F$ increases computational cost.

Figure 3 shows learning curves of ablated versions of MAT against the transfer attack. In accordance with Table 1, training with a larger number of meta-perturbations $P$, more iterations in I-FGSM, and with a REPTILE learning rate smaller than 1.0 consistently improves robustness.

While Table 1 shows the worst accuracy of a setting against all attacks of the grid search, Figure 4 summarizes the accuracy of all attacks of the grid search in a box plot. Each value is averaged over 5

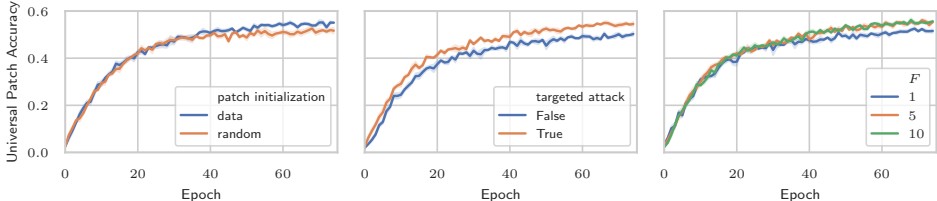

Figure 2: Comparison of MAT models' learning curves against the transfer attack for generating universal patches. (Left) meta-perturbation initialization in MAT between *data initialization* and *random initialization*. (Middle) MAT using targeted attacks and untargeted attacks for updating meta-perturbations. (Right) MAT uses the worst meta-perturbation chosen from different numbers of samples $F$.

independent runs with 5 different seeds for each training procedure. Each model is evaluated against three patch attack procedures: *data*, *random*, and *low frequency*. Configurations with large variance indicate that the model might appear to be robust if hyperparameters of the attack are chosen badly. This effect is particularly pronounced for AT and SAT against the random initialized S-PGD attack, where only very few attack configurations are able to strongly degrade performance.

Moreover, the results exhibit that MAT is the only model robust against the data initialization attacks. None of the attacks reduce MAT's accuracy below 0.5 regardless of initialization methods. As discussed before, attacks through data initialization are more effective than through random initialization and attacks employing a low frequency filter are most effective on MAT with random initialization (MATr). Nevertheless, MATr still shows stronger robustness than all other approaches except MAT.

### C.1.2 Robustness against Universal Perturbation Attacks

We present an analogous evaluation as in Section 4.2 for a universal perturbation attack. We use the same dataset, neural architecture, and training pipeline but train the models specifically for universal perturbation attacks. We allow universal perturbations $\xi$ with $||\xi||_\infty \leq 20/255 \approx 0.078$. In comparison with training models against universal patch attacks, the key difference is that the models are trained specifically for universal perturbation attacks instead of universal patch attacks. Following the training configuration shown in Table 7, we train ResNet V1 models with 4 training approaches - Standard, AT, UAT, and MAT. We do not present results for SAT (Mummadi et al., 2019) since we have not found a stable configuration of hyperparameters for this setting; however, the results of Mummadi et al. (2019) indicate that SAT should perform slightly better than AT when configured appropriately. We evaluate the robustness of the models against the same attacks as for patch attacks.

| SETTING | AT | UAT | MAT |
|---|---|---|---|
| PATCH INITIALIZATION | RANDOM | RANDOM | RANDOM |
| WORST OVER $F$ SAMPLES | 1 | 1 | 5 |
| REPTILE LEARNING RATE $\sigma$ | 0 | 1 | 0.25 |
| NUMBER OF META-PERTURBATIONS $P$ | – | 1 | 1000 |
| I-FGSM STEP-SIZE $\alpha$ | [0.0001, 0.02] | 0.01 | [0.0001, 0.02] |
| I-FGSM ITERATIONS $K$ | 5 | 1 | 5 |

Table 7: Configuration of MAT with different approaches against universal perturbation attacks for image classification.

The evaluation results are summarized in Table 8. In comparison with the results of universal patch attacks in Table 1, we notice a few interesting differences: firstly, clean accuracy is degraded for all variants of adversarial training compared to standard training. This indicates a trade-off between clean performance and robustness in this threat-model. Secondly, in contrast to standard training, AT and UAT made non-trivial gains in robustness, whereas their robustness did not improve against universal patches addressed in Section 4.2. Thirdly, the accuracy of UAT in Table 8 shows that UAT

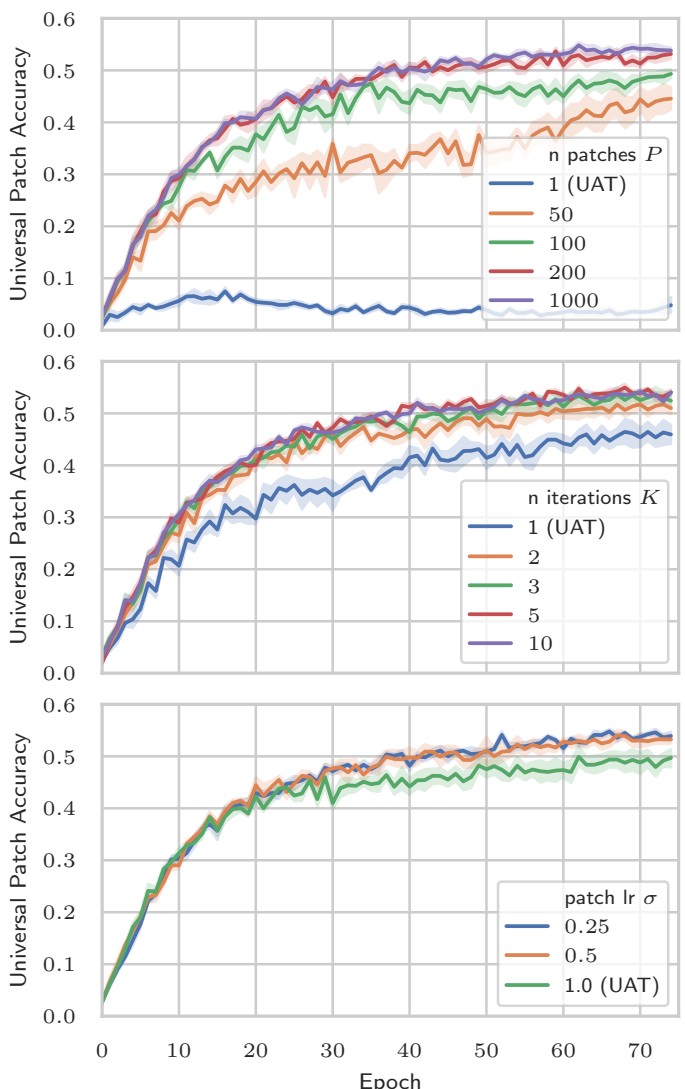

Figure 3: Ablation study of MAT in the aspects where it differs from UAT (Shafahi et al., 2018). Shown is accuracy on Tiny ImageNet against a patch generated with the transfer attack: UAT learns a single perturbation/patch while MAT learns an ensemble (upper plot). UAT performs one iteration of I-FGSM while MAT performs multiple (middle). UAT uses joint training ($\sigma = 1$), while MAT uses full REPTILE with $\sigma \leq 1$ (bottom plot).

| SETTING | CL | RI | DI | LF | TR | MIN |
|---|---|---|---|---|---|---|
| STANDARD | **0.55** | 0.03 | 0.04 | 0.03 | 0.03 | 0.03 |
| AT (MADRY ET AL., 2018) | 0.48 | 0.33 | 0.26 | 0.33 | 0.27 | 0.21 |
| UAT (SHAFAHI ET AL., 2018) | 0.46 | 0.23 | 0.23 | 0.23 | 0.17 | 0.17 |
| MAT (FULL) | 0.48 | **0.42** | **0.42** | **0.42** | **0.39** | **0.39** |

Table 8: Accuracy (mean over 5 runs) of different methods against universal perturbation attacks on Tiny ImageNet.

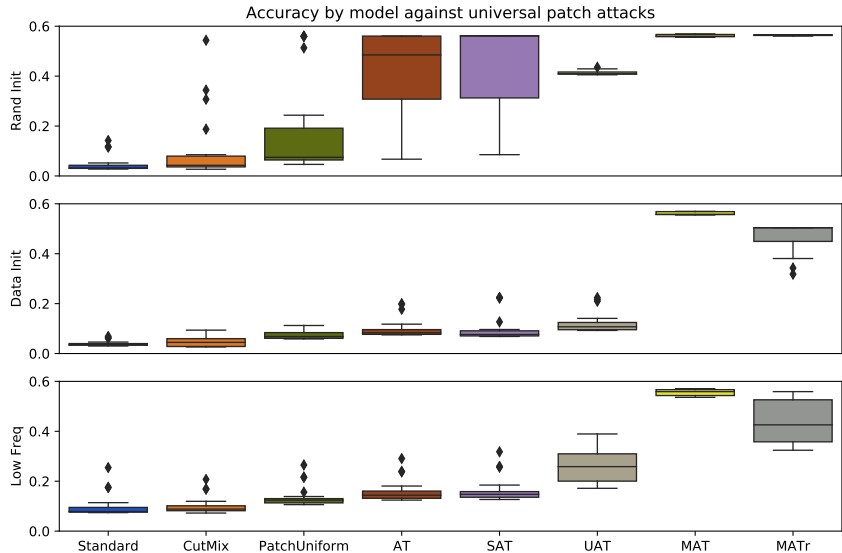

Figure 4: Robustness against universal patch attacks: results correspond to Table 1 but show distribution of accuracy over elements of the grid search rather than only the worst accuracy. Each value is averaged across 5 runs (with 5 different seeds) for each configuration per training approach. MATr is the MAT model trained with randomly initialized meta-perturbations, whereas MAT represents the model trained with meta-perturbations through data initialization. The rows correspond to three different universal patch attacks.

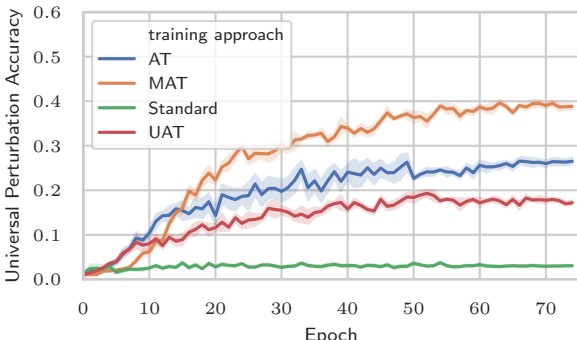

Figure 5: Comparison of learning curves of Standard training, AT, UAT, and MAT against the transfer attack for generating universal perturbations.

overfits less strongly to the randomly initialized S-PGD attack compared to the universal patch attacks in Table 1. Despite these differences, MAT considerably outperforms all other methods in terms of robustness also in this setting.

Figure 5 shows the learning curves of those training approaches against the transfer attack for generating universal perturbations. Notably, while MAT is less robust in the early phase of training, it reaches a significantly higher level of robustness in the end.

Figure 6 shows the box plot corresponding to Table 8. The accuracy of MAT models is above 0.4 for all three attacks and shows little variance. In contrast, UAT and AT are robust against certain attack configurations but against an optimally configured attack, accuracy degrades to 0.25 or less. This shows that evaluating robustness reliably requires a strong set of attacks and their well-tuned hyperparameters.

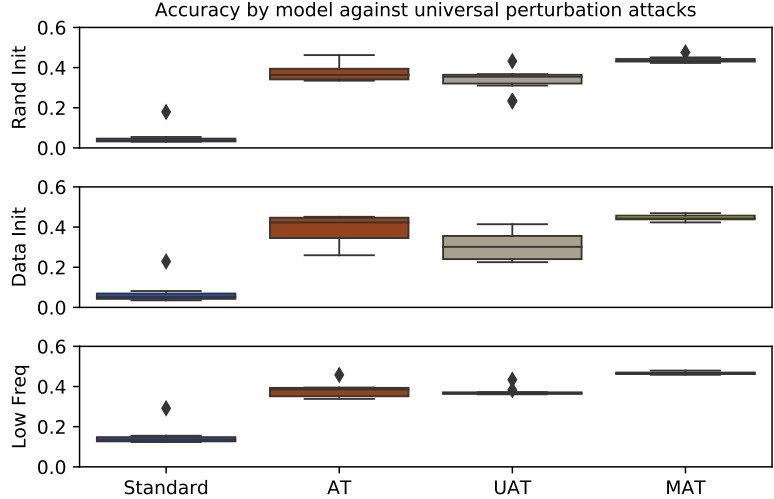

Figure 6: Robustness against universal perturbation attacks: results correspond to Table 8 but show distribution of accuracy over elements of the grid search rather than only the worst accuracy. Each value is an average across 5 runs (with 5 different seeds) for each configuration per training approach. The rows correspond to three different universal patch attacks.

## C.2 OBJECT DETECTION ON BOSCH SMALL TRAFFIC LIGHTS DATASET

We show the box plots corresponding to Table 2 in Figure 7 for recall and mAP, respectively. Notably, only the recall of the standard model can be reduced considerably (meaning true positives can be hidden) and this requires an appropriately configured attack. Interestingly, a low-frequency attack is not effective for reducing the recall of any model. In contrast, low-frequency attacks are the most effective ones for reducing mAP, that is: for causing false positive detections. While randomly initialized S-PGD is not successful at reducing the mAP of any model besides the standard model, many low-frequency attacks of varied attack configurations reduce mAP of most models (except MAT + data) considerably. In contrast, S-PGD from data initialization can be effective but fails in most cases to reduce mAP for all but the standard model.

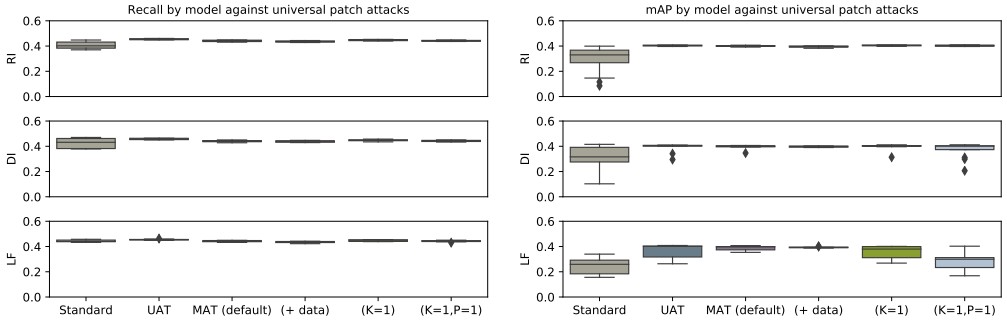

Figure 7: Recall (left) and mean Average Precision (right) by model against universal patch attacks on Bosch Small Traffic Lights Dataset. MAT (default): meta perturbation is initialized uniform-randomly. MAT (+data): meta perturbation is initialized from a cropped image. Both MAT (default) and MAT (+data) have I-FGSM iteration $K=3$ and number of patches $P = 10$. MAT($K$=1): I-FGSM iteration $K$=1 and number of patches $P = 10$. MAT($K = 1, P = 1$): I-FGSM iteration $K$=1 and number of patches $P = 1$.

## D    ILLUSTRATION OF PATCH ATTACKS ON TINY IMAGENET

We illustrate universal patch attacks on models trained on Tiny ImageNet in Figure 8. Note that these are the strongest patches found against these models during the grid search. Oftentimes, the generated patch resembles the target class: examples for this are the low-frequency attack on a standard model (fooling it to mistake a chimpanzee for a police van), the random initialization attack against the SAT model (fooling it to mistake the chimpanzee for a ladybug), the data initialization attack against the SAT model (fooling it to mistake the chimpanzee for an orange), or the low-frequency attack against the UAT model (fooling it into classifying the input as a fire salamander based on the characteristic texture of the patch). While these misclassifications can be explained, a human would very likely still classify the inputs as chimpanzees. Attacks on MAT (full) fail to generate interpretable patches; however, transferring patches generated for other models (such as the shown ones) to MAT does not cause misclassifications either.

## E    ILLUSTRATION OF PATCH ATTACKS ON BOSCH SMALL TRAFFIC LIGHTS DATASET

We illustrate universal patch attacks on models trained on Bosch Small Traffic Lights Dataset in Figure 9. Note that these are the strongest patches found against these models during the grid search in terms of the mAP. These patches often invoke high confidence false detections. However, MAT with data initialization does not show any false positives.

For the patches found for MAT (Data Init), we show the progress of the patches during the attack in Figure 10. Similarly, Figure 11 shows the patches' evolution during an attack on the standard model. Note that patches converge fairly quickly, namely, running attacks longer would not make them stronger. Moreover, all three patches for MAT converge to a red-cyan pattern and the patches for data and random initialization exhibit very similar patterns. This indicates that this pattern is actually a minimizer of the loss with a large basin of attraction. However, as Figure 9 shows, it does not really fool the model. Finally, Figure 12 shows the training of the patch shown in Figure 1.

## F    ILLUSTRATION OF PERTURBATION ATTACKS ON TINY IMAGENET

We illustrate universal perturbation attacks on models trained on Tiny ImageNet in Figure 13. Note that these are the strongest perturbations found against these models during the grid search.

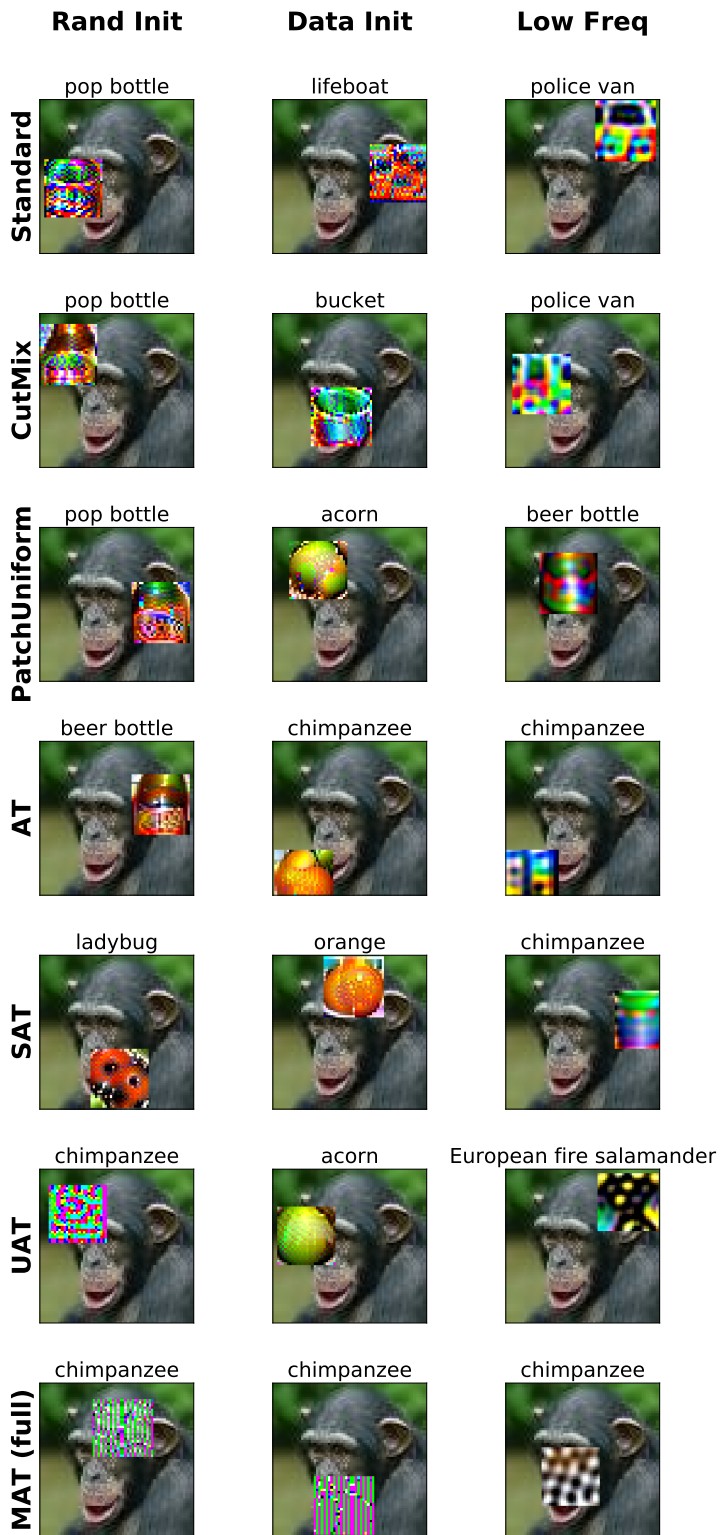

Figure 8: Best random initialization, data initialization, and low-frequency attacks against the respective models from Table 1. The model prediction is given above each plot. The correct label is *chimpanzee*.

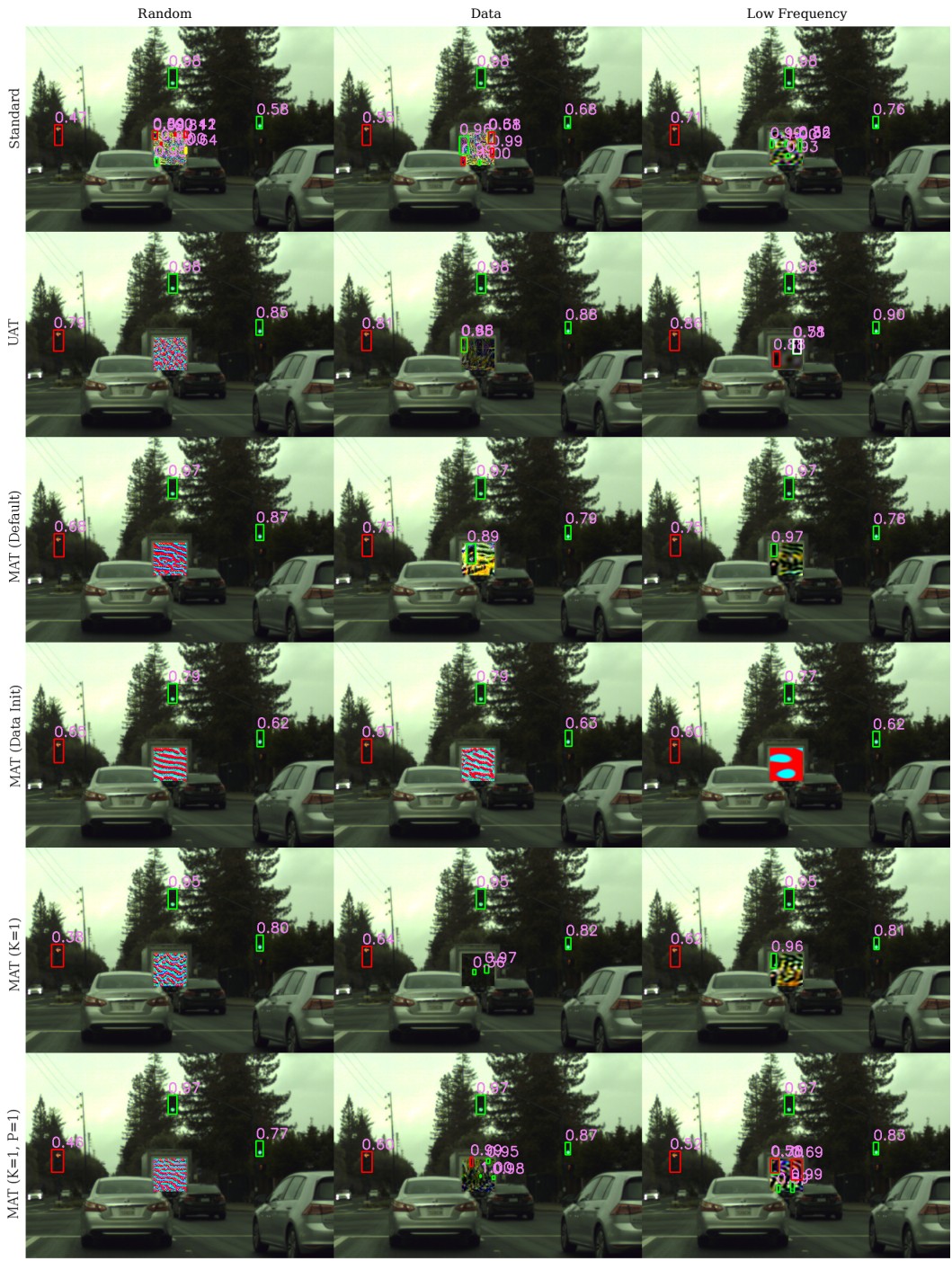

Figure 9: Best (in terms of mAP) random initialization, data-crop initialization and low-frequency attacks against the models from Table 2. The patch location is fixed for a better comparison.

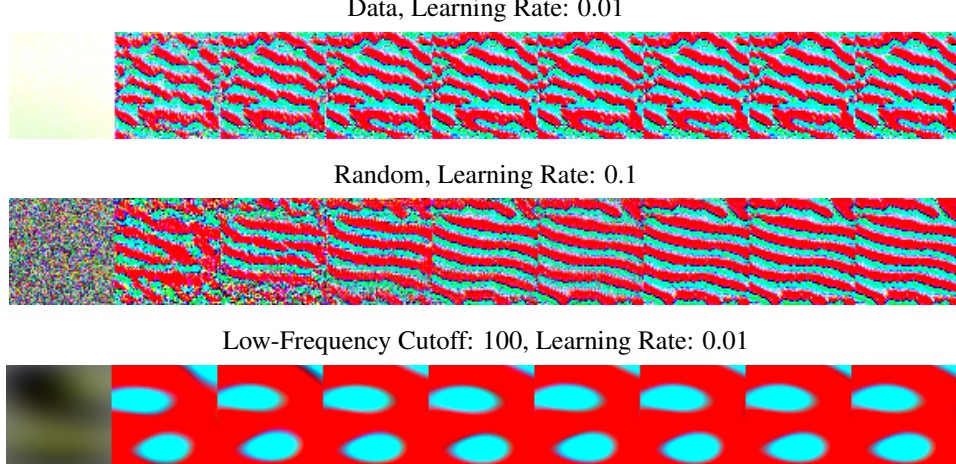

Figure 10: Training of patches against a MAT (Data Init) model. Attack from data-crop initialization with step size of 0.01 (top). Attack from random initialization with step size of 0.1 (center). Low-frequency attack with cutoff frequency 100 from data-crop initialization with step size of 0.001.

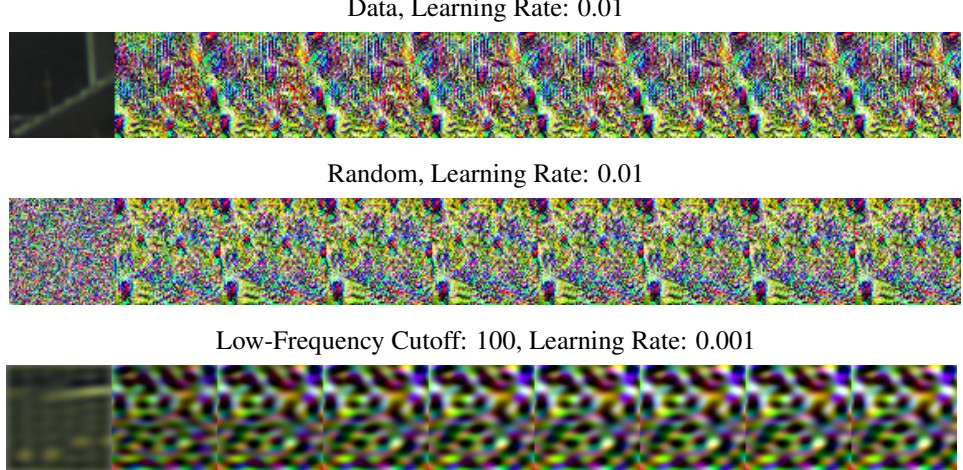

Figure 11: Training of patches against standard model: Attack from data-crop initialization with step size of 0.01 (top). Attack from random initialization with step size of 0.01 (center). Low-frequency attack with cutoff frequency 100 from data-crop initialization with step size of 0.001

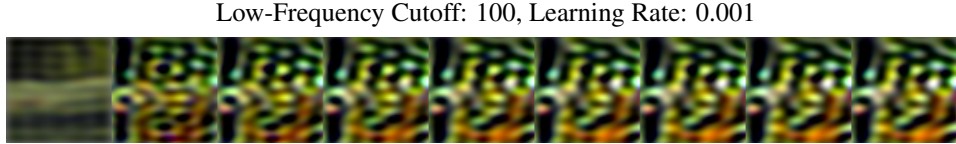

Figure 12: Training of patch from Figure 1. This patch is generated with a low-frequency attack with cut-off frequency 100, learning rate of 0.001 and starting from data-crop initialization.

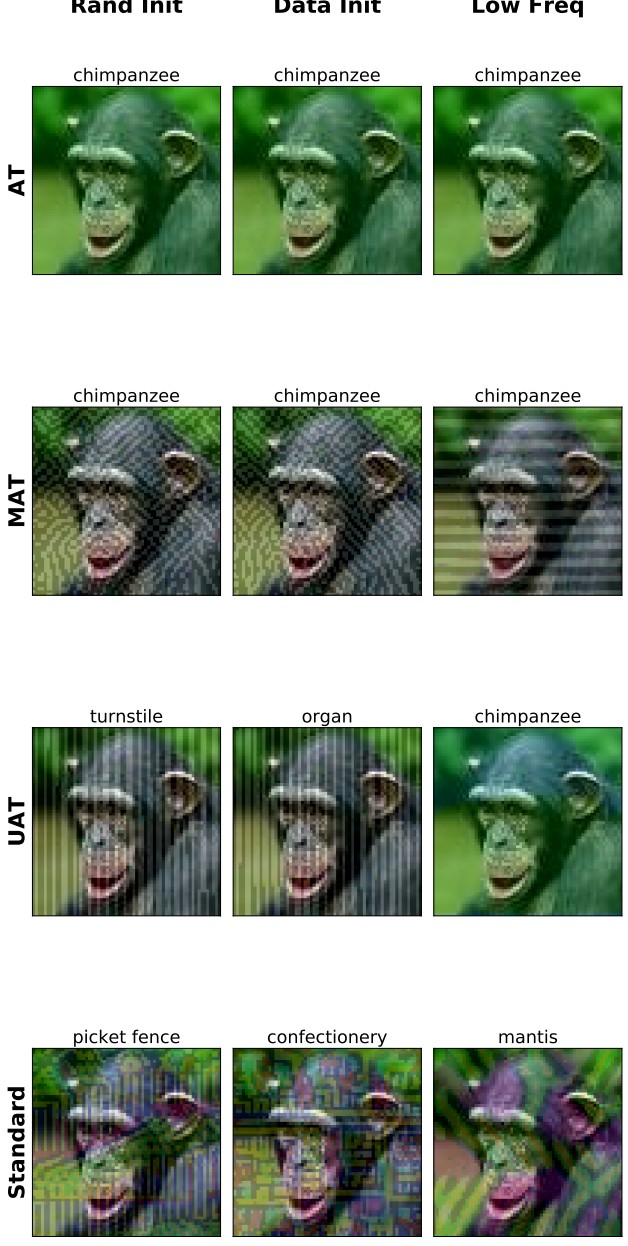

Figure 13: Best random initialization, data initialization, and low-frequency perturbation attacks against the respective models from Table 8. The model prediction is given above each plot. The correct label is *chimpanzee*.

