# OpenReview forum: "Meta Adversarial Training"
_ICLR.cc/2021/Conference — Reject_

### Official Review · AnonReviewer2 · 2020-10-29
**Good results on defending against universal patch attacks, but clarity of writing should be improved**

**Rating:** 6
**Confidence:** 4

**Review:**

This paper studies the use of meta learning and adversarial training to defend against universal perturbations. This approach tries to learn a set of perturbations through meta-learning and train the model to defend against such attacks. Experimental results on Tiny ImageNet and the Bosch small traffic lights dataset show the method is effective in defending against universal patch-based attacks.

Although the main idea of the paper is sound and empirical evaluations promising, there are some clarity issues with the writing of the paper:
1. In Algorithm 1, important operations such as INIT, SELECT, and UPDATE are described informally in the text instead of defined properly.
2. What is the attack method used in Table 1? The columns only describe the initialization.
3. Is the AT model in Table 1 trained using patch-based attacks or usual whole image attack?
4. The notations used in Section 3.2 is for \xi is very confusing and too overloaded. What do the notations \xi(\theta_t_1), \xi(\theta_t_i)_{i=1}^N, \xi_t^m, \xi_{T=k}, \xi_{t=K} mean? The authors are using notations liberally without definitions.
5. What does S-PGD stand for?

Also, in learning meta-perturbations the authors use different step sizes \alpha in I-FGSM and targeted attacks from different classes for a diverse collection of attacks. The same idea of diversifying the attacks can also be incorporated in AT, SAT or UAT training. Have the authors tried this when performing comparisons?

Overall I think this paper is a solid piece of work, but the clarity of presentation has to be improved.

---

> ### Author Response · Authors · 2020-11-13
> **Clarifying questions of the reviewer**
>
> We would like to thank the reviewer for the thorough review and the constructive feedback. We will try to clarify the open points (and revise the paper accordingly):
> 1. "In Algorithm 1, important operations such as INIT, SELECT, and UPDATE are described informally in the text instead of defined properly."
> We will define INIT, SELECT,and UPDATE more properly, thanks for the suggestion.
>
> 2. "What is the attack method used in Table 1? The columns only describe the initialization."
> The attack method used in Table 1 is S-PGD (see point 5). More details are given in Appendix A (in the supplementary material). We will move Appendix A to the main document.
>
> 3. "Is the AT model in Table 1 trained using patch-based attacks or usual whole image attack?"
> The AT model (and SAT and UAT as well) in Table 1 are trained using patch-based attacks.
>
> 4. "The notations used in Section 3.2 is for \xi is very confusing and too overloaded. What do the notations \xi(\theta_t_1), \xi(\theta_t_i){i=1}^N, \xi_t^m, \xi_{T=k}, \xi{t=K} mean? The authors are using notations liberally without definitions."
> We will reduce complexity of the notation and get rid of the \xi(\theta) part. The remaining notations are as follows: Upper-case T indexes the time step of the inner maximization (I-FGSM), lower-case t indexes the time step of the REPTILE meta-learner, and m indexes the meta-perturbation. We will try to make this more clear in the paper.  UPDATE:  We have now simplified notation for perturbations as follows: upper-case $\Xi$ for meta-perturbations, lower-case $\xi$ for perturbations, subscript $_t$ for indexing outer minimization, and superscript $^{(k)}$ for indexing inner maximization.
>
> 5. "What does S-PGD stand for?"
> S-PGD is the stochastic mini-batch version of PGD (just as SGD is the stochastic version of gradient descent). S-PGD is formally summarized in Appendix A. To provide a short informal summary:
> For universal perturbations/patches, the perturbation needs to be optimized for maximizing the expected loss with the expectation being over the data distribution (and randomness of the perturbation/patch application), see Equation 1. S-PGD samples a mini-batch from the available data, computes the gradient with respect to the loss on the mini-batch (rather than on the full data as PGD would do), and performs a single step of PGD. In the next step of S-PGD, a different mini-batch is sampled.
>
> 6. "Also, in learning meta-perturbations the authors use different step sizes \alpha in I-FGSM and targeted attacks from different classes for a diverse collection of attacks. The same idea of diversifying the attacks can also be incorporated in AT, SAT or UAT training. Have the authors tried this when performing comparisons?"
> Regarding different step sizes $\alpha$ in AT, SAT, UAT: thanks for bringing this up; in fact, we sample the step size $\alpha$ in AT as well. Details on this can be found in Appendix B.3. We repeat this part here and hope that it clarifies the reviewer's point: "The most crucial parameter for AT, SAT, and UAT is the step size $\alpha$ of I-FGSM. For AT, we follow MAT and sample the step size per datapoint randomly from a log-uniform distribution over $[0.0001, 0.1]$. Since UAT and SAT only update a single patch per batch, this random sampling strategy is not feasible on a per-batch level. Instead, we use a fixed $\alpha$; more specifically, we use 0.2 for SAT such that I-FGSM can reach any value in $[0, 1]^d$ in K=5 iterations. Since UAT updates the perturbation iteratively over the batches, a smaller value for $\alpha$ is feasible here and we employ $\alpha=0.01$. We did not tune these choices for $\alpha$ extensively but note that MAT does not require any tuning of the step size $\alpha$."
>
> We will shortly upload a revised version of the paper. In case our response leaves questions open, we would be grateful if the reviewer could discuss this with us further.

---

### Official Review · AnonReviewer1 · 2020-10-29
**An interesting work MAT (by coupling adversarial training and meta-learning).**

**Rating:** 6
**Confidence:** 4

**Review:**

The authors propose a novel meta adversarial training method. In particular, to improve the robustness against digital domain attacks, the proposed meta adversarial training (MAT) combines adversarial training and meta-learning, which reduces the computing cost compared with adversarial training by generating a set of stronger perturbations. The proposed approach sounds with extensive experimental results.

* Pros:

1.	The authors study a crucial issue of adversarial attacks, namely, a robust defense to perturbations. This problem is practical and of great importance to many areas, such as autonomous driving, robotics, etc.

2.	The proposed MAT is designed for robust defense by combining adversarial training and meta-learning.

3.	In MAT, meta-learning is used to meta-learn a set of initial perturbations and followed by a few gradient steps for adaptations. This method can avoid overfitting and the generated perturbations will try to focus on diverse weaknesses of the model, and thus the model is more robust.

4.	The authors conducted comprehensive experiments, including image classification and object detection, with both qualitative analysis and quantitative results provided to show the effectiveness of the proposed model.

* Cons:

1.	In meta-training process, REPTILE is used to update the model $\cal M$. However, it seems that in the testing stage, $\cal M$ is directly used rather than adapted using training data from the testing task. As a result, the learned model $\cal M$ is more like a pretrained model, rather than meta learning. It would nice if the authors can explain the details of testing process, and the design choice.

2.	In Algorithm 1, it would be nice to explain in more details of the choices of INIT and SELECT functions, and their impacts to the algorithm performances.

---

> ### Author Response · Authors · 2020-11-13
> **Clarifying the reviewer's concerns**
>
> We would like to thank the reviewer for the time devoted to reviewing the paper and the constructive feedback. We will try to clarify the open points:
>
> Con 1: "In meta-training process, REPTILE is used to update the model M. ..."
>  We would like to clarify that we do not meta-learn weights of the model M. We do meta-learn initializations for the training-time attack (meta-perturbations/meta-patches). In terms of the robustness evaluation, our paper follows standard procedures: model weights are fixed (no test-time adaptation is conducted) and attacks are performed independently of the meta-learned perturbations (see Appendix A for details).
>
> Con 2: " In Algorithm 1, it would be nice to explain in more details of the choices of INIT and SELECT functions, and their impacts to the algorithm performances."
> We will define INIT and SELECT more formally in a revised version of the paper. In terms of their impact on algorithm performance, ablations for those functions are already contained in Table 1 and in Figure 2 in the appendix (we will make this more evident in the paper):
>   * (Random Init) The INIT function initializes the set of meta perturbations. The two options to implement this function are random initialization and on-manifold/data initialization. Ablation experiments with random initialization show that it degrades robustness, which justifies our decision to use on-manifold initialization as default.
>   * (Untargeted) corresponds to an ablation of MAT where INIT does not assign fixed target classes to meta-perturbations (such that they get optimized in an untargeted way). This also impairs robustness, which justifies our decision to use targeted attacks that increase diversity of meta-perturbations.
>   * (F=1) corresponds to an ablated version of MAT where SELECT uniform randomly picks a meta-perturbation (and corresponding target and step size) from $\mathcal{P}$ rather than taking the most effective out of F=5 samples. F=1 again deteriorates robustness, which validates our choice of the proposed SELECT with F=5. (F=10 does not further increase robustness, see Figure 2 in the appendix)
>   * The number of meta-perturbations/patches P created in INIT is studied in Figure 3 in the appendix: ablating MAT by decreasing the number of meta-perturbation degrades robustness.
>
> We will shortly upload a revised version of the paper. In case our response leaves questions open, we would be grateful for discussing these further.

---

### Official Review · AnonReviewer5 · 2020-11-05
**Review of the paper "meta adversarial training"**

**Rating:** 5
**Confidence:** 4

**Review:**

#########################################################################

Summary:
This paper uses meta learning to learn adversarial perturbations to improve the model robustness against universal patch attacks.

#########################################################################

Pros:
1. The proposed meta adversarial training (MAT) claims to learn a large & diverse collection of universal perturbations that aids the robust training.
Specifically, this paper leverages the “reptile” meta learning method to learn the better initial values of universal perturbations, which may lead to better universal perturbations for updating the model.
The meta learner can be trained in parallel to the robust training.

2. The writing is sound. The logic flow is clear.

#########################################################################

Cons:
1 This paper mentioned existing defenses overfit universal perturbation. Would the proposed method MAT relieve this overfitting issue?

2 How does the adversarial training (e.g., against l_p norm bounded attack) resist the universal perturbation, compared with specially designed defenses such as MAT?
In table 1, why AT cannot defense transfer attack? Could it be possible that the baseline methods are improperly configured?

3 How & why does the initialization of the universal perturbations matters most? (Since the author claims the meta learning can better help the initialization.)
To be specific,  are there any justifications & explanations on the better initialization for aiding robustness?

4 Another drawback of this paper is the unclear/incomplete technical descriptions (including Section A.)
#########################################################################

---

> ### Author Response · Authors · 2020-11-13
> **Addressing the concerns**
>
> We would like to thank the reviewer for the helpful review and the provided suggestions. We will try to clarify the open points:
> Con 1:  "This paper mentioned existing defenses overfit universal perturbation. Would the proposed method MAT relieve this overfitting issue?"
> With "overfitting" in this context, we refer to: a model is robust against specific universal perturbations or patches that are used or generated at training time, but not against arbitrary universal perturbations or patches generated by other attacks at test time. For instance, UAT in Table 1 and ablated MAT with "Random Init" have high accuracy against "Random Init" attacks (used during training) but low accuracy against stronger attacks. Since MAT (full) has high accuracy against a broad range of attacks, including ones never used during training, we are confident that the model would relieve this issue.
>
> Con 2: "How does the adversarial training (e.g., against l_p norm bounded attack) resist the universal perturbation, compared with specially designed defenses such as MAT? In table 1, why AT cannot defense transfer attack? Could it be possible that the baseline methods are improperly configured?"
> Adversarial training (AT) was not trained against l_p norm bounded attacks but against patch attacks. We discuss how hyperparameters of baselines such as step sizes are chosen in Section C.1; we would also like to note that MAT does not require any explicit tuning of the step size. Therefore, we can confidently say our results are not due to the baselines being "improperly configured". We attribute the subpar performance of the baselines to the observation that "initialization matters" for patch attacks (see below). MAT explicitly addresses this initialization problem. As mentioned before, AT does not defend against transfer attacks due to overfitting, while it is robust against the attacks used during training (S-PGD with random init).
>
> Con 3: "How & why does the initialization of the universal perturbations matters most? (Since the author claims the meta learning can better help the initialization.) To be specific, are there any justifications & explanations on the better initialization for aiding robustness?"
> There is strong empirical evidence that the initialization of the universal patches matters: in fact, the core difference between AT and MAT is the initialization of the perturbation for inner maximization (uniform random for AT, meta-learned for MAT). The strong gap in robustness in Table 1 can thus be attributed to this key difference. To explain this for the specific case of universal patches: uniform random initialized patches have very different statistics than actual image patches. A model under adversarial training can learn to mostly ignore "unnatural/high frequency patches" such as uniform random patches. This in turn means that there is no useful gradient information for uniform random patches pointing towards more effective patches - in the most extreme case a uniform random patch is ignored entirely by the model, its gradient is zero, and I-FGSM based optimization of the patch fails accordingly. In contrast, initial patches discovered through meta learning typically appear more "natural" to the model, contribute already to its decision, and accordingly have more useful gradient information pointing towards even more effective patches.
>
> Con 4: "Another drawback of this paper is the unclear/incomplete technical descriptions (including Section A.) "
> Could the reviewer detail in which parts the technical description is unclear/incomplete? Is there anything beyond the points raised by the other reviewers?
>
> We will shortly upload a revised version of the paper. In case our response leaves questions open, we would be grateful if the reviewer could discuss this with us further.

---

### Author Response · Authors · 2020-11-19
**Revised version of submission**

We have uploaded a revised version of our submission based on the comments by the reviewers. We would like to thank all reviewers again for their valuable input! Below we list the main changes compared to the initial submission:
* Section on reliable robustness evaluation with different variants of S-PGD moved from appendix to main document as Section 4.1, where we also clarified the details of S-PGD (AnonReviewer2, AnonReviewer5)
 * Simplify notation for perturbations (AnonReviewer2). We use now consistently the following notation: upper-case $\Xi$ for meta-perturbations, lower-case $\xi$ for perturbations, subscript $t$ for indexing outer minimization, and superscript $^{(k)}$ for indexing inner maximization.
 * Added pseudo-code for INIT and SELECT as Algorithm 2 and 3 in appendix (AnonReviewer1, AnonReviewer2)
 * Algorithm 1 and Section 3.4 are completely revised such that they are more accessible and clarity is improved (AnonReviewer5)
 * Clarify that all baseline defenses were trained against patch attacks (AnonReviewer5, AnonReviewer2)
 * Clarify that we do not meta-learn model weights (AnonReviewer1)
 * Please note that we have moved the appendix from the supplementary material to the main document such that hyperlinks work.

We believe these changes improve the quality of the submission and would like to thank the reviewers again for their valuable suggestions and comments.

Since the main criticism of the reviewers was in terms of clarity and we believe these points have been addressed appropriately, we would be very grateful if the reviewers would consider our revised submission for the final review and decision.

If any points remain open, we would be happy for discussing them further.

We thank the reviewers for the time devoted to reviewing our submission.

---

### Decision · Program_Chairs · 2021-01-07
**Final Decision**

**Decision:**

Reject

**Comment:**

I thank the authors and reviewers for the discussions. Reviewers agreed the work is interesting but there are some aspects of the paper that need improvements. In particular, authors need to better address concerns raised by R5. Given all, I think the work still needs a bit more work before being accepted.